# Policy changes and the screening, diagnosis and treatment of drug-resistant tuberculosis patients from 2015 to 2018 in Zhejiang Province, China: a retrospective cohort study

Weixi Jiang [ORCID] ,[1] Ying Peng,[2] Xiaomeng Wang,[2] Chris Elbers,[3] Shenglan Tang,[4] Fei Huang,[5] Bin Chen,[2] Frank Cobelens[6]

For numbered affiliations see end of article.

**Correspondence to**
Mr Bin Chen; bchen@cdc.zj.cn

## ABSTRACT

**Objectives** To examine changes in the screening, diagnosis, treatment and management of drug-resistant tuberculosis (DRTB) patients, and investigate the impacts of DRTB-related policies on patients of different demographic and socioeconomic characteristics.

**Design** A retrospective cohort study using registry data, plus a survey on DRTB-related policies.

**Setting** All prefecture-level Centres for Disease Control in Zhejiang Province, China.

**Main outcome measures** Alongside the care cascade, we examined: (1) reported number of presumptive DRTB patients; (2) percentage of presumptive patients with drug susceptibility testing (DST) records; (3) percentage of DRTB/rifampicin-resistant (RR) patients registered; (4) percentage of RR/multidrug-resistant TB (MDRTB) patients that received anti-DRTB treatment; and (5) percentage of RR/MDRTB patients cured/completed treatment among those treated. Multivariate logistic regressions were conducted to explore the impacts of DRTB policies after adjusting for other factors.

**Results** The number of reported presumptive DRTB patients and the percentage with DST records largely increased during 2015–2018, and the percentage of registered patients who received anti-DRTB treatment also increased from 59.0% to 86.5%. Patients under the policies of equipping GeneXpert plus expanded criteria for DST had a higher likelihood of being registered compared with no GeneXpert (adjusted OR (aOR)=2.57, 95% CI: 1.20 to 5.51), while for treatment initiation the association was only significant when further expanding the registration criteria (aOR=2.38, 95% CI: 1.19 to 4.79). Patients with registered residence inside Zhejiang were more likely to be registered (aOR=1.96, 95% CI: 1.52 to 2.52), treated (aOR=3.83, 95% CI: 2.78 to 5.28) and complete treatment (aOR=1.92, 95% CI: 1.03 to 3.59) compared with those outside.

**Conclusion** The policy changes on DST and registration have effectively improved DRTB case finding and care. Nevertheless, challenges remain in servicing vulnerable groups such as migrants and improving equity in the access to TB care. Future policies should provide comprehensive support for migrants to complete treatment at their current place of residence.

### Strengths and limitations of this study

► This study conducted a comprehensive and systematic cascade analysis on the healthcare pathway for drug-resistant tuberculosis (DRTB) patients: screening, diagnosis, treatment and management, and examined key factors associated with the case finding and healthcare process.

► This study evaluated the effectiveness of different combinations of DRTB-related policies as they were implemented in all prefectures of one Chinese province, including new diagnostic technology promotion, improved patient management and financial support.

► This study also examined those patients left-behind during the policy changes from an equity perspective, with a focus on rifampin monoresistant patients and patients with registered residence (Hukou) outside Zhejiang.

► The impacts of the financing policies could not be explored as almost all prefectures had some form of financing policies since 2015 and there were no individual-level data on whether the patient benefited from these financing policies.

► Other potentially important socioeconomic factors like education and income were not available in the dataset used, limiting, therefore, the scope of the equity analysis to migrant workers.

## INTRODUCTION

Drug-resistant tuberculosis (DRTB) has become a global concern in recent years. It is estimated that globally in 2018, 3.4% of new TB patients and 18% of previously treated patients developed rifampicin-resistant (RR) forms of disease, including multidrug-resistant TB (MDRTB), in which there is additional resistance to isoniazid, and 7.1% and 21%, respectively, in China.[1] Currently, China accounts for 14% of the RR/MDRTB disease burden, and one recent projection suggested

the incidence of RR/MDRTB would triple without interventions to change current conditions.[2] The RR/MDRTB epidemic has posed a great challenge to achieve the target of Sustainable Development Goals (SDGs) of ending TB in 2030 in China.[3]

The diagnosis and treatment of DRTB can be very costly in terms of both time and money,[4] and barriers to accessing DRTB diagnosis and care exist worldwide.[5] The diagnosis of DRTB takes 1–3 months using traditional technology, and the treatment of RR/MDRTB lasts for up to 2 years. DRTB patients generally have lower socioeconomic status, and the cost of treatment is so high that the current financing policies in the form of health insurance reimbursement and subsidised treatment are far from sufficient.[6–8] For China specifically, the public health insurance programmes provide very limited coverage for outpatient services that are required for around 20 months for RR/MDRTB patients. In addition, some second-line anti-TB drugs and auxiliary drugs are often not covered.[9–11] Previous studies in several countries, including China, have also revealed long treatment delay, high pre-diagnosis and pre-treatment attrition, and high loss-to-follow-up during the treatment course.[12–17] Moreover, the MDRTB treatment success rate in China was less than 50%.[18]

These formidable barriers for DRTB patients to accessing and adhering to standard treatment call for strong supporting policies for patients to receive and complete treatment.[19] Previous studies have validated the utility of rapid drug susceptibility testing (DST) technologies such as Genechip and GeneXpert in the screening for DRTB, and shown that these technologies could improve DRTB case finding, shorten treatment delays and decrease pre-diagnosis attrition.[13 20–24] As for treatment adherence, one comprehensive programme in China that provided universal health coverage to MDRTB patients was shown to improve access to and affordability of diagnosis and quality treatment of MDRTB.[5] Studies on other intervention strategies showed that directly observed therapy or other reminding approaches through digital technologies could improve the treatment adherence and outcomes of TB/MDRTB.[25 26] Patient counselling alone or combined with financial support can also increase the cure rates among MDRTB patients.[27]

In China, some provinces, along with or after the roll-out of international donor-funded projects, have implemented policies to improve DRTB control, including allocating special funds to equip DRTB designated hospitals with DST facilities and reagents, improving health insurance benefit packages and providing subsidies to patients.[11 28] However, there is limited evidence on the impact of implementing these policies on DRTB case finding and care thereafter. The equity issues underlying the case finding and care procedures for DRTB patients are also understudied, especially considering that the eligibility for policies issued in a certain region is often linked with patients' registered residence, work and health insurance status.

This study aims to examine changes in the programmatic performance with regard to screening, diagnosis, treatment and management of DRTB patients in Zhejiang from 2015 to 2018 through a cascade analysis approach.[29] In addition, as the policies and guidelines on this whole procedure of DRTB care changed during this period, this study systematically summarises these changes and investigates how these policy changes have influenced case finding and treatment of DRTB patients. We also explore whether these policy changes have equally influenced patients of different demographic and socioeconomic characteristics.

## METHODS
### Study settings
Zhejiang is a province located in the eastern area with its Gross Domestic Product (GDP) ranking fourth in China,[30] and has a growing migrant population in recent years. Under the current TB control model in Zhejiang, the prefecture-level designated hospital, usually one in each prefecture, is responsible for the diagnosis and treatment of DRTB patients, while the Centers for Disease Control and Prevention (CDC) and primary healthcare facilities conduct patient management. Patients who are clinically suspected to have DRTB are to be referred for DST and should be reported in the Tuberculosis Information Management System (TBIMS) as presumptive DRTB patients. Their sputum samples are to be sent to the prefecture-level designated hospital for DST. If diagnosed with DRTB, these patients should be registered in the TBIMS database for diagnosed patients as such. The criteria regarding what types of DRTB should be registered have being changing over time.

### Study design
This study included a questionnaire survey on the DRTB policies/programmes among the 11 prefecture-level CDCs and a quantitative analysis of the TBIMS records of presumptive and diagnosed DRTB patients.

### Questionnaire survey on DRTB policies/programmes
A questionnaire on the DRTB-related policies was distributed to the CDC of all prefectures in Zhejiang in collaboration with Zhejiang provincial CDC. After preliminary consultation with the provincial CDC, the questionnaire was designed to include policies in four areas: (1) eligibility of presumptive DRTB patients for DST; (2) eligibility for registering DRTB patients based on the type of drug resistance (including rifampin monoresistance (RMR), MDR, extensive drug resistance (XDR) and monoresistance to other types of drugs); (3) DST technology and payment for DST and (4) financing policies, including both health insurance benefit packages and government subsidies for supporting the DRTB treatment. Detailed questions on the eligibility for financial support regarding the types of DRTB, the registered permanent residence and the region of health insurance enrolment were also included

in the questionnaire (see online supplemental material 1). If the policies had changed at any time after 2015, we collected information on the policy details before and after the change as well as the year of change.

## TBIMS records

De-identified TBIMS records of presumptive and diagnosed DRTB patients from 2015 to 2018 in Zhejiang were retrieved from the National Center for Tuberculosis Control and Prevention of China, CDC. The dataset of presumptive DRTB patients included demographic information (prefecture of registration, age, sex, ethnicity, occupation and registered residence), drug-resistance test profile (the date of sending the sample, conducting the test and reporting the result, the type of test and the test result), as well as a unique registration number if registered as a DRTB patient in the diagnosed patient dataset. Only those DRTB patients recorded in the diagnosed patient dataset have a traceable treatment history and are managed under a specialised guideline for DRTB patients. The dataset of diagnosed DRTB patients contained the same demographic information and diagnostic data as the dataset of presumptive patients, plus the treatment information, including the starting date of anti-DRTB treatment, including both second-line (RR/MDRTB) and adapted first-line (monoresistance to isoniazid) treatment, TB treatment history, current state of treatment (under treatment or ended) and the ending date of treatment as well as the reason of ending the treatment (cured/treatment completed/death/lost-to-follow-up/others) if ended. In TBIMS, the treatment status referred to whether the patient received treatment in Zhejiang. If patients were registered but returned to their hometown for treatment, it could be shown as 'treatment refused'.

## Patient and public involvement statement

Patients or the public were not involved in the design, or conduct, or reporting, or dissemination plans of our research.

## Data analysis

Using datasets of the presumptive and registered diagnosed patients, a seven-step cascade of diagnosis and care was constructed for each year from 2015 to 2018 : (1) reported presumptive DRTB patients; (2) presumptive patients who had DST; (3) diagnosed DRTB, and RR/MDRTB (including RMR, MDR and XDR) patients; (4) registered RR/MDRTB patients; (5) RR/MDRTB patients that ever received anti-DRTB treatment; (6) RR/MDRTB patients that were under treatment 6 months after treatment initiated; and (7) RR/MDRTB patients that had been cured or completed treatment, as shown in figure 1. Diagnosed DRTB patients were defined as patients with a DST result showing resistance to at least one type of first-line anti-TB drug. As in 10 out of the 11 prefectures only RR/MDRTB patients were required to be registered and thus had available data on treatment history before 2019, we only analysed RR/MDRTB patients for steps 4–7. Descriptive analyses were conducted to explore the reasons for attrition at each step based on the relevant records in the datasets. Patients' ethnicity was categorised as Han and other minority groups. Frequencies of patient characteristics were also calculated across patient groups at each step of the cascade to examine potential factors associated with the attrition. For steps 3–5, the frequencies of some patient characteristics of interests, including age, sex, drug-resistant type, occupation and registered residence, were calculated separately for each year to examine the trends from 2015 to 2018.

Mixed-effect two-level logistic regressions, with fixed slope and random intercept specification, were conducted to explore factors associated with the likelihood of being registered for RRTB patients, the likelihood of receiving anti-DRTB treatment if registered and the likelihood of getting cured or completing treatment (the latter only for those who initiated treatment before 2017 because of the 2-year treatment duration). This model showed better fit to the data than the fixed-effect model based on the likelihood-ratio test. The DRTB-related policies were summarised along the four areas in the questionnaires. Besides patient-level variables, the policies implemented in each prefecture and each year, categorised according to the different combinations of policies in the four areas, were included in the model. Per-capita GDP of the prefecture was divided into three groups, as the numbers assembled in three intervals: RMB 55000–70

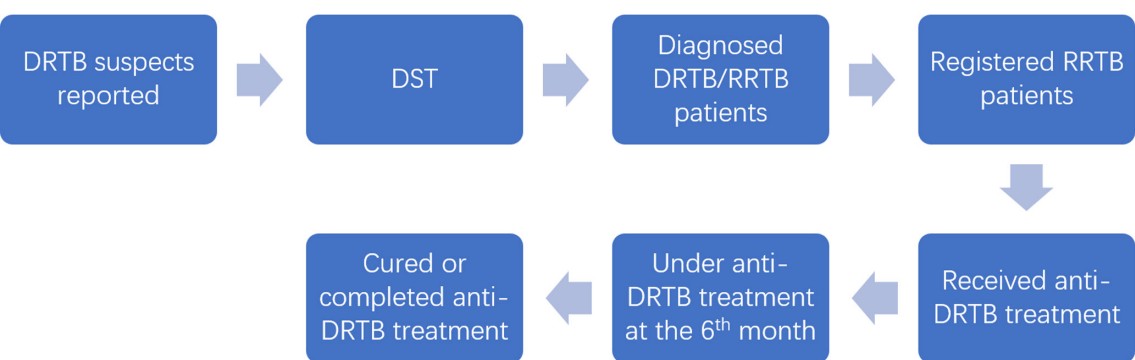

**Figure 1** Seven steps of the DRTB diagnosis and care cascade. DRTB, drug-resistant tuberculosis; RRTB, rifampicin-resistant tuberculosis.

000, RMB 80 000–100 000 and above RMB 120 000, and also included in the model as a categorical variable. Standard errors were estimated accounting for clustering, considering the intra-prefecture correlations.

## RESULTS

### DRTB policy change from 2015 to 2018 in Zhejiang

Results from the questionnaire survey on DRTB policies showed that the major change in the eligibility of patients for DST was the expansion from high-risk patients (including smear positive relapse, treatment failure and positive sputum bacteriology after 2 months' treatment), to all bacteriologically positive TB patients. As for the testing technology, the major change was the introduction of the rapid DST assay GeneXpert MTB/RIF that combines testing for *Mycobacterium tuberculosis* with screening for rifampicin-resistance with a same-day result, while the costs for the test were all covered, or mostly covered by the government. The criteria for DRTB patient registration (implying eligibility for specialised DRTB patient management) also expanded from MDR/XDRTB to RRTB, and one prefecture even expanded to any DRTB before 2018. In most prefectures, these changes in DST eligibility and equipment as well as the registration criteria happened in 2017, while almost all others had adopted the new polices and introduced GeneXpert earlier (online supplemental table 1).

The financing policies to support DRTB treatment in Zhejiang consisted of both health insurance and medical assistance policies, and in most prefectures, these policies had been implemented in 2015. These financing policies aimed to improve the benefit package for DRTB patients through a three-layer coverage system: (1) increasing the reimbursement rate of outpatient treatment to 70%–90%, the same level as for inpatient treatment; (2) the out-of-pocket (OOP) expenditure exceeding a certain amount, typically RMB 20 000–30 000, could be further reimbursed at a rate of 55%–85% through a supplemental health insurance programme for critical illnesses; (3) the OOP expenditure after health insurance reimbursement could be covered by the medical assistance, with a payment limit varying from RMB 11 000 to over RMB 60 000 across prefectures. As for the eligibility for the insurance policies, those with public health insurance enrolled

outside Zhejiang, usually migrants without formal jobs, were excluded. In 5 out of the 11 prefectures, RMR patients were still not eligible for the expanded public health insurance benefit package up to 2019. For the medical assistance, in 6 out of the 11 prefectures, only patients with registered residence ('Hukou') in Zhejiang were eligible, and for the other 4 prefectures with assistance policies, the eligibility had been expanded to all residents in this prefecture. Up to 2018, RMR patients in five prefectures were still excluded from the medical assistance (online supplemental table 2).

### Cascade analysis of DRTB care from 2015 to 2018

Table 1 shows the screening and diagnosis of DRTB patients from 2015 to 2018. In general, the results suggested increased capacity to find DRTB patients. The number of reported presumptive DRTB patients increased from 9285 to 23 916, with the largest increase from 2016 to 2017 coinciding with the change in screening policy in most prefectures. The percentage of patients with test results also increased from 69.3% to 78.1%. Along with the increase in the volume of DST, there was an increase in the number of diagnosed DRTB patients except for 2018, and a decrease in the percentage of patients diagnosed among those tested. RR/MDRTB patients accounted for around 40%–50% of the total DRTB patients.

Further analysis on the types of diagnostic tests patients received revealed an increasing trend in the percentage of patients taking rapid DST as well as the percentage of patients tested with both conventional culture-based and rapid DST. The percentage of presumptive patients without DST profile for unknown reasons as recorded in the TBIMS dataset, which indicates pre-diagnosis attrition, dropped dramatically from 17.3% to 2.7% (online supplemental table 3).

Table 2 shows the registration and treatment for diagnosed RRTB patients from 2015 to 2018. While the actual number of registered RRTB patients increased, the percentage of registered RRTB patients dropped in 2017, and this percentage increased again to 84.6%. The percentage of registered patients that received anti-DRTB treatment increased from 59.0% to 86.5%, and the percentage of those treated who received at least 6-month treatment remained above 90%. Around 70% of patients diagnosed in 2015 and 2016 completed treatment.

**Table 1** Diagnostic cascade starting from presumptive DRTB patients

| Year | Number of presumptive DRTB patients | Patients with test records | | Diagnosed DRTB patients | | Diagnosed RR/MDRTB patients | |
|---|---|---|---|---|---|---|---|
| | | Number | % | Number | % | Number | % |
| 2015 | 9285 | 6434 | 69.3 | 1031 | 16.0 | 503 | 48.8 |
| 2016 | 10 997 | 8438 | 76.7 | 1258 | 14.9 | 529 | 42.1 |
| 2017 | 21 768 | 14 764 | 67.8 | 1729 | 11.7 | 716 | 41.4 |
| 2018 | 23 916 | 18 670 | 78.1 | 1580 | 8.5 | 663 | 42.0 |

DRTB, drug-resistant tuberculosis; MDRTB, multidrug-resistant tuberculosis; RR, rifampicin resistant.

**Table 2** Registration and treatment cascade for diagnosed RRTB patients

| Year | Diagnosed RR/ MDRTB patients | Registered | | Received anti-DRTB treatment | | Under treatment 6 months after treatment initiated | | Cured or completed treatment | |
|---|---|---|---|---|---|---|---|---|---|
| | | No. | % | No. | % | No. | % | No. | % |
| 2015 | 503 | 383 | 76.1 | 226 | 59.0 | 215 | 95.1 | 158 | 73.5 |
| 2016 | 529 | 410 | 77.5 | 283 | 69.0 | 263 | 92.9 | 182 | 69.2 |
| 2017 | 716 | 502 | 70.1 | 360 | 71.7 | 334 | 92.8 | – | – |
| 2018 | 663 | 561 | 84.6 | 485 | 86.5 | 462 | 95.3 | – | – |

DRTB, drug-resistant tuberculosis; MDRTB, multidrug-resistant tuberculosis; RRTB, rifampicin-resistant tuberculosis.

Duration of treatment could be longer than 2 years, as 38 out of the 283 patients starting treatment in 2016 were still shown as under treatment at the time we retrieved the data (see online supplemental table 4).

### Factors associated with the registration and treatment of DRTB patients

Table 3 shows the characteristics of patients across each step of the care cascade. Around 70% of the presumptive and diagnosed patients were male, and around 98% of them were Han People. Only around 40% of the patients had formal jobs other than farming or unemployment. While under the assumption of perfect equity we would expect that the percentage of patients with different characteristics remained the same from the diagnosed to the treated group, it decreased from 30.3% to 27.0% for older patients, 28.1% to 22.5% for RMR patients, and

**Table 3** Patient characteristics across each step of the care cascade

| | Presumptive DRTB patients | Diagnosed DRTB patients | Diagnosed RR/ MDRTB patients | Registered RR/ MDRTB patients | RR/MDRTB patients received treatment |
|---|---|---|---|---|---|
| N | 65966 | 5598 | 2411 | 1859 | 1357 |
| Age (%) | | | | | |
| ≥60 | 41.3 | 35.9 | 30.3 | 29.8 | 27.0 |
| Gender (%) | | | | | |
| Male | 70.6 | 73.6 | 72.5 | 72.7 | 72.7 |
| Ethnicity | | | | | |
| Han (%) | 97.7 | 98.0 | 97.9 | 98.0 | 98.8 |
| Patient type (%) | | | | | |
| New patient | 70.2 | 65.2 | 51.8 | 51.4 | 48.8 |
| Drug-resistant type (%) | | | | | |
| RMR | – | 12.1 | 28.1 | 20.8 | 22.5 |
| MDR | – | 30.4 | 70.6 | 78.0 | 76.1 |
| XDR | – | 0.5 | 1.2 | 1.2 | 1.4 |
| Non-rifampicin resistant | – | 56.9 | – | – | – |
| Registered residence (%) | | | | | |
| Outside Zhejiang | 23.4 | 25.2 | 24.4 | 22.3 | 17.0 |
| Job category (%) | | | | | |
| Farmers | 49.0 | 48.0 | 43.8 | 45.0 | 45.5 |
| Unemployed | 12.4 | 13.0 | 14.1 | 14.2 | 14.0 |
| Other | 38.6 | 39.0 | 42.1 | 40.8 | 40.5 |
| Test type (%) | | | | | |
| Fast test | – | 38.6 | 46.5 | 46.1 | 49.2 |

DRTB, drug-resistant tuberculosis; MDRTB, multidrug-resistant tuberculosis; RMR, rifampin monoresistant; RR, rifamipicin resistant; XDR, extensive drug resistance.

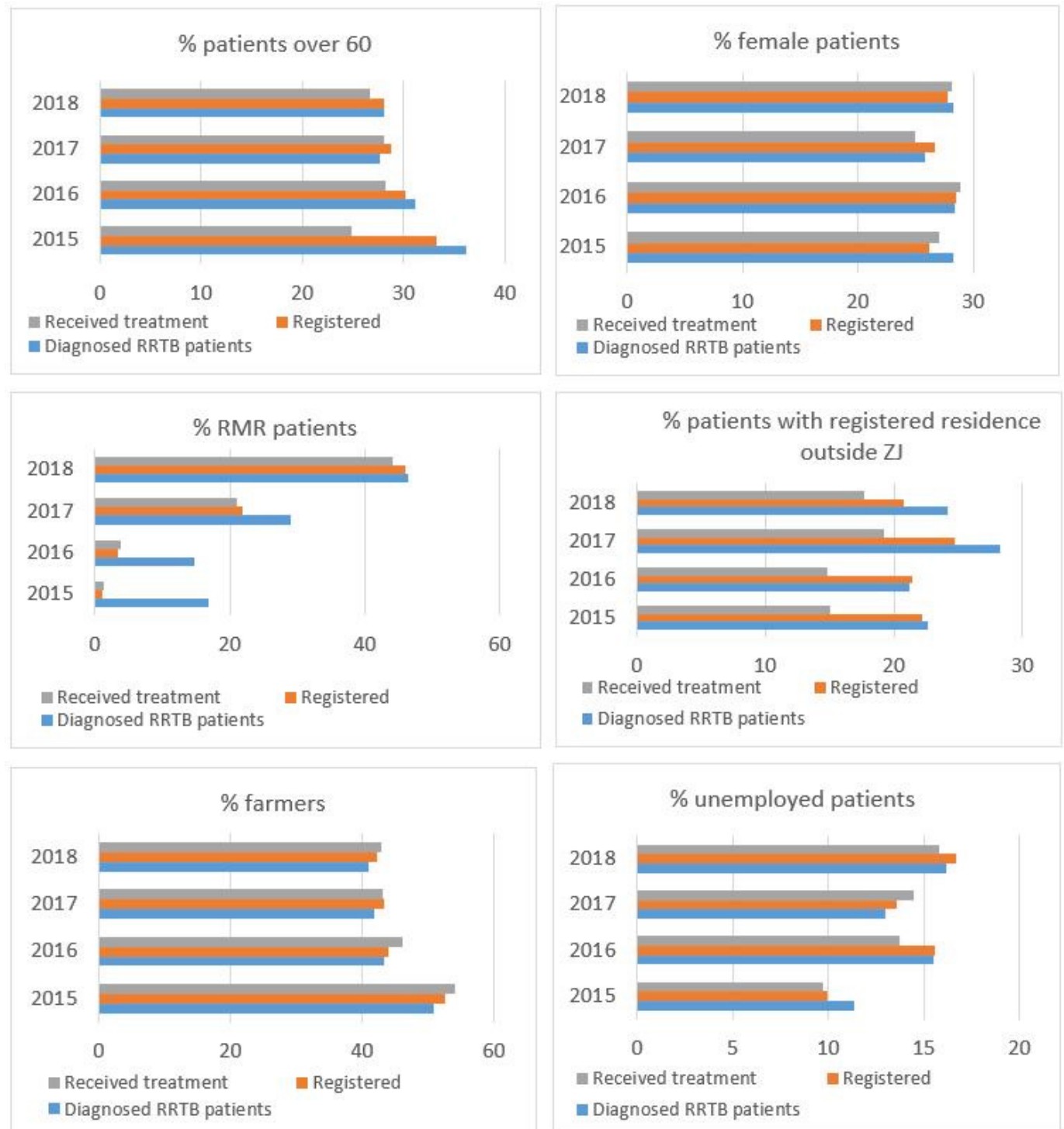

**Figure 2** Characteristics of diagnosed, registered and treated RR/MDRTB patients, 2015–2018. RMR, rifampin monoresistant; RRTB, rifamipicin-resistant tuberculosis; ZJ, Zhejiang.

from 24.4% to 17.0% for patients with registered residence outside Zhejiang, that is, migrants. An increase was observed regarding the percentage of patients ever taking rapid DST.

Figure 2 shows the changes in the characteristics of diagnosed, registered and treated RR/MDRTB patients from 2015 to 2018. The percentage of diagnosed RMR patients that received adequate treatment was low in 2015 and

2016, but increased dramatically from 2017, coinciding with the policy change. The gaps between the proportion of older patients in the diagnosed and the treated group also narrowed. Nevertheless, over the 4 years, there remained a 7%–10% decrease in the proportion of patients with registered residence outside Zhejiang from the diagnosed to the treated patient group, indicating that migrants were still more likely to drop out after

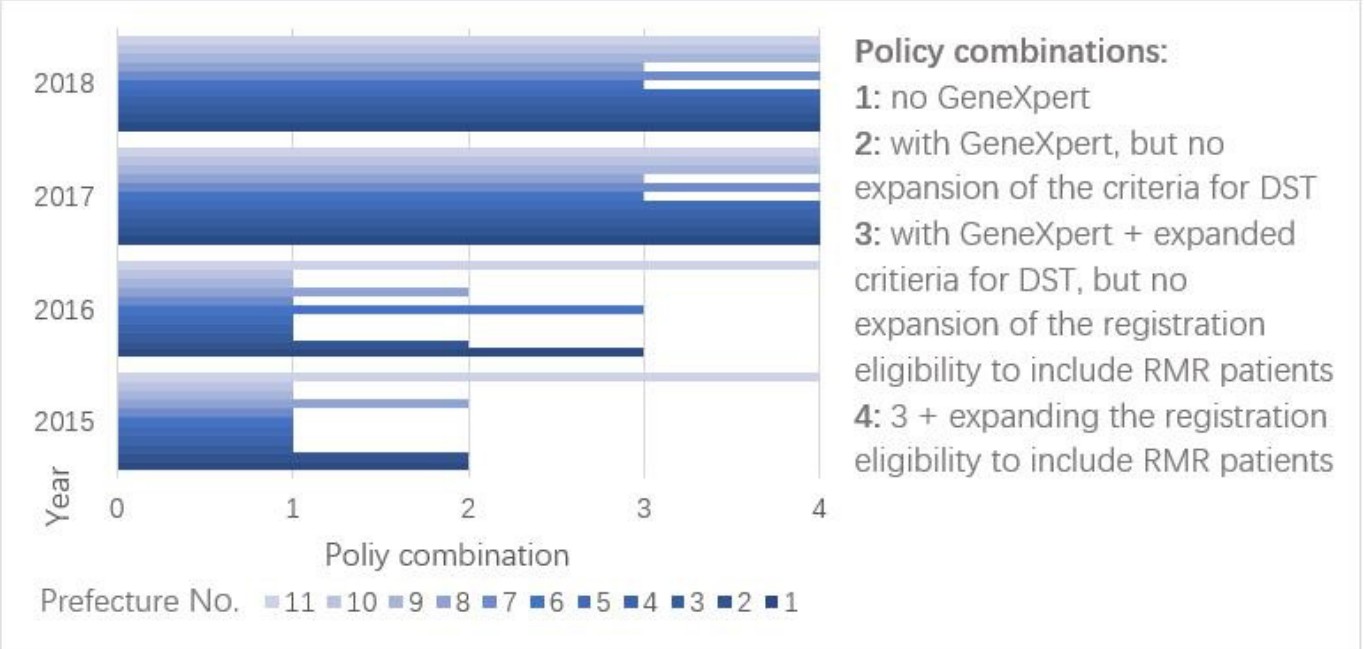

**Figure 3** Combination of DRTB policies implemented in each prefecture from 2015 to 2018. DST, drug susceptibility testing; RMR, rifampin monoresistant.

diagnosis compared with the local residents. Gender and job category were not obviously associated with the pretreatment attrition in this univariate analysis.

To systematically investigate the impact of policies on DRTB case finding and care, we divided the combination of policies in different areas of DRTB implemented in each prefecture for each year during 2015–2018 into four categories, considering both policy implications and the number of cases in each category. As shown in figure 3, since 2017, all prefectures had equipped GeneXpert and expanded the eligibility criteria for DST as well as registration, except for two prefectures which still did not register RMR patients.

Table 4 shows the results of mixed-effect two-level logistic regression of the factors associated with registration among diagnosed RRTB patients, as well as with receiving anti-DRTB treatment among those registered. After adjusting for other factors, patients under the policy of providing GeneXpert together with expanding eligibility for DST (category 3, adjusted OR (aOR)=2.57, 95% CI: 1.20 to 5.51) had a significantly higher likelihood of being registered compared with patients not provided with GeneXpert, while this association was not significant for providing GeneXpert testing without expanding eligibility (category 2). A positive association with receiving anti-DRTB treatment was significant only for the policy combination of providing GeneXpert and expanding eligibility for both DST and registration (category 4, aOR=2.38, 95% CI: 1.19 to 4.79). Not surprisingly, MDR/XDRTB patients were much more likely to be registered compared with RMR patients due to the registration policy in earlier time periods. In addition, patients with registered residence inside Zhejiang were more likely

to be registered (aOR=1.96, 95% CI: 1.52 to 2.52) or treated (aOR=3.83, 95% CI: 2.78 to 5.28). Older age was associated with lower likelihood both of being registered (aOR=0.69, 95% CI: 0.53 to 0.90) and of receiving anti-DRTB treatment (aOR=0.40, 95% CI: 0.30 to 0.52).

Table 5 shows the results of the mixed-effect two-level logistic regression analysis of factors associated with the likelihood of completing treatment for patients who initiated treatment before 2017. All prefectures had some kind of financing policies in 2015 and 2016, and most prefectures had not yet expanded the eligibility of patients for DST or registration. We therefore used the individual level data on utilisation of rapid DST to assess the impact of rapid testing on treatment completion. After adjusting for other factors, older patients were less likely to complete treatment (aOR=0.24, p<0.001), while registered residence inside Zhejiang was associated with higher probability of completing treatment (aOR=1.92, p=0.04). These two factors had significant impacts of the same direction on the registration, treatment initiation and treatment completion of RR/MDRTB patients. Utilisation of rapid DST was not associated with treatment completion in this study.

## DISCUSSION
Results from this study clearly revealed the positive impacts of the combined policy changes regarding DST of presumptive DRTB patients and registration for proper management of diagnosed patients. However, inequity challenges remain in terms of servicing vulnerable groups, for example, migrant workers and the older population, in the registration, treatment and management of DRTB.

**Table 4** Factors associated with registration for diagnosed RRTB patients, and with receiving anti-DRTB treatment for registered RRTB patients

| | Registration (n=2367) | | | | Receiving treatment (n=1824) | | | |
|---|---|---|---|---|---|---|---|---|
| | OR | P>z | 95% CI | | OR | P>z | 95% CI | |
| Age | | | | | | | | |
| ≥60 | 0.69 | 0.006 | 0.53 | 0.90 | 0.40 | 0.000 | 0.30 | 0.52 |
| Gender | | | | | | | | |
| Male | 1.08 | 0.554 | 0.85 | 1.37 | 0.87 | 0.425 | 0.62 | 1.22 |
| DR type | | | | | | | | |
| MDR/XDRTB | 5.93 | 0.000 | 3.10 | 11.36 | 0.91 | 0.789 | 0.45 | 1.84 |
| Patient type | | | | | | | | |
| New patient | 1.22 | 0.146 | 0.93 | 1.59 | 0.52 | 0.000 | 0.37 | 0.72 |
| Treatment history | NA | | | | | | | |
| No treatment | | | | | ref. | | | |
| First-line drug only | | | | | 0.58 | 0.044 | 0.34 | 0.99 |
| Second-line drug used | | | | | 1.11 | 0.748 | 0.58 | 2.12 |
| Policy category | | | | | | | | |
| 1 | ref. | | | | ref. | | | |
| 2 | 1.18 | 0.626 | 0.61 | 2.29 | 0.87 | 0.776 | 0.34 | 2.25 |
| 3 | 2.57 | 0.015 | 1.20 | 5.51 | 1.44 | 0.257 | 0.77 | 2.68 |
| 4 | 2.08 | 0.054 | 0.99 | 4.37 | 2.38 | 0.015 | 1.19 | 4.79 |
| Registered residence | | | | | | | | |
| In Zhejiang | 1.96 | 0.000 | 1.52 | 2.52 | 3.83 | 0.000 | 2.78 | 5.28 |
| Job category | | | | | | | | |
| Farmers | ref. | | | | ref. | | | |
| Unemployed | 1.21 | 0.017 | 1.04 | 1.42 | 0.90 | 0.651 | 0.58 | 1.40 |
| Other jobs | 0.92 | 0.281 | 0.80 | 1.07 | 0.94 | 0.784 | 0.60 | 1.47 |
| Per capita GDP | | | | | | | | |
| Highest group | ref. | | | | ref. | | | |
| Middle group | 1.72 | 0.122 | 0.87 | 3.42 | 2.20 | 0.025 | 1.10 | 4.38 |
| Lowest group | 1.70 | 0.099 | 0.90 | 3.20 | 1.15 | 0.643 | 0.63 | 2.12 |
| _cons | 0.32 | 0.002 | 0.16 | 0.66 | 1.85 | 0.207 | 0.71 | 4.78 |
| prefecture var(_cons) | 0.18 | | 0.09 | 0.39 | 0.18 | | 0.03 | 0.92 |

DRTB, drug-resistant tuberculosis; GDP, Gross Domestic Product
; MDR, multidrug resistant; RRTB, rifampicin-resistant tuberculosis; XDRTB, extensive drug resistance tuberculosis.

### Effectiveness and challenges of the policy interventions

The expansion of the eligibility criteria of presumptive DRTB patients referred for DST, together with the increased funding support to equip the facilities and guarantee the supply of reagents for rapid DST like GeneXpert, has greatly improved the capacity for DRTB case finding. The significant effects of the combined policies compared with providing GeneXpert alone indicates that in order to effectively improve case finding we need to not just introduce new technologies, but also support and expand their use. The inclusion of RMR in the registration and management of DRTB patients closed the management gap between RMR and MDR/XDRTB patients, and also improved the treatment rate for RR/MDRTB.

It is obvious that these reforms in the DRTB policies and regulations have effectively changed the practices in DRTB control. Nevertheless, no policies aiming at migrants were issued between 2015 and 2018, and during these 4 years the degree of inequity between local and migrant patients did not seem to be mitigated either. The improved insurance benefit package was only available for those with public health insurance enrolment in Zhejiang, and in many prefectures receiving medical assistance still required local registered residence during 2015–2018. In addition, older people were less likely to

**Table 5** Factors associated with the likelihood of completing treatment for patients that initiated treatment before 2017

| | OR | P value | 95% CI | Interval |
|---|---|---|---|---|
| **Age** | | | | |
| ≥60 | 0.24 | <0.001 | 0.14 | 0.42 |
| **Gender** | | | | |
| Male | 0.79 | 0.442 | 0.44 | 1.44 |
| **Drug-resistance type** | | | | |
| MDR/XDRTB | 0.50 | 0.086 | 0.23 | 1.10 |
| **Patient type** | | | | |
| New patient | 1.06 | 0.820 | 0.64 | 1.77 |
| **TB treatment history** | | | | |
| No treatment history | Ref. | | | |
| First-line drug only | 0.74 | 0.355 | 0.39 | 1.40 |
| Second-line drug used | 1.07 | 0.847 | 0.56 | 2.02 |
| **Test type** | | | | |
| Fast test | 0.70 | 0.096 | 0.46 | 1.06 |
| **Registered residence** | | | | |
| In Zhejiang | 1.92 | 0.040 | 1.03 | 3.59 |
| **Job category** | | | | |
| Farmers | Ref. | | | |
| Unemployed | 0.84 | 0.635 | 0.42 | 1.71 |
| Other jobs | 1.44 | 0.278 | 0.74 | 2.79 |
| **Per capita GDP** | | | | |
| Highest group | Ref. | | | |
| Middle group | 0.94 | 0.939 | 0.20 | 4.39 |
| Lowest group | 1.07 | 0.921 | 0.30 | 3.74 |
| **Year of sending sample** | | | | |
| 2015 | Ref. | | | |
| 2016 | 1.23 | 0.164 | 0.92 | 1.64 |
| _cons | 5.58 | 0.001 | 2.01 | 15.50 |
| prefecture var(_cons) | 0.21 | | 0.04 | 1.10 |

MDR, multidrug resistant; XDRTB, extensive drug-resistance tuberculosis.

be registered or receive treatment. This may be because they were more likely to give up treatment due to the high cost and long course of the standard treatment, and health workers may not register them once they refused to provide information necessary for registration. All these findings were consistent with previous studies that age and migration for work, as well as health system factors such as lack of clear eligibility criteria for DST and limited capacity to provide DST were associated with attrition at different stages of the cascade.[14 16 31–33] Nevertheless, no impacts were observed of other factors, such as association between treatment experience and treatment outcome, and data on other socioeconomic factors such as financial difficulties were not available in this study.

### Equity challenges regarding migrant populations
The growing number of migrants in China and other parts of the world are posing a challenge to TB control.[31 34–36]

Although one systematic review found no significant differences in treatment adherence between migrant and long-term resident MDRTB patients,[37] several studies identified barriers for migrants to accessing TB diagnosis and care mainly in terms of knowledge gaps and financial difficulty,[38 39] and these barriers would likely to be more substantial for migrant MDRTB patients. In our study, only around 40% of the DRTB patients had a formal job other than farmers or self-employment. In China, people with a formal job will be compulsively enrolled in the urban employee basic medical insurance at the place of work as required by law. However, migrants without a formal job often choose to participate in the health insurance for residents in their hometown, which requires lower premium compared with the more developed region they migrated to, and thus cannot benefit from the reimbursement policy in their working place. Besides, they often

have to pay the full cost of treatment first before they get reimbursed when they return home. Therefore, it is expected that many of these DRTB patients would choose to go back to their hometown for treatment, or even refuse treatment. The need for migrant DRTB patients to travel long distances and the substandard or even absent treatment received would increase the risk of disease transmission and treatment failure. A study in Shanghai, a well-developed city in eastern China, showed that financial incentives were effective for migrant TB patients to complete treatment.[40] In terms of these findings, policies aimed at encouraging migrant DRTB patients to be treated and managed at their current place of residence need to be developed, and this is particularly urgent for regions with a large migrant population.

## Limitations

One major limitation of our study is that almost all prefectures had some form of financing policy since 2015 and there were no individual-level data on whether the patient benefited from these financing policies. Therefore, in this study, the impacts of these financing policies could not be explored. Furthermore, other potentially important socioeconomic factors like education and income were not available in our dataset, limiting the scope of the equity analysis. Nevertheless, these limitations would not influence our conclusions that changes in the screening and registration policies of DRTB patients have largely increased the case finding and management capacity of DRTB patients, while during the period 2015–2018 differences between the local and migrant patients in the registration, treatment initiation and treatment outcome remained. We did not consider population changes either as the resident population only increased 5% from 2015 to 2018 in Zhejiang.[41] Future research should collect more individual-level data on the implementation of the health insurance and medical assistance policies to investigate the impacts of these demand-side policies on DRTB treatment more directly, and explore what kind of financing policies provided for migrants could increase their likelihood of initiating and completing anti-DRTB treatment.

**Author affiliations**
¹Global Health Research Center, Duke Kunshan University, Kunshan, China
²Zhejiang Provincial Center for Disease Control and Prevention, Hangzhou, China
³School of Business and Economics, Vrije Universiteit Amsterdam, Amsterdam, The Netherlands
⁴Duke Global Health Institute, Duke University, Durham, North Carolina, USA
⁵National Center for Tuberculosis Control and Prevention, Centers for Disease Control and Prevention, Beijing, China
⁶Department of Global Health, Amsterdam Institute for Global Health and Development, Amsterdam University Medical Centres, Duivendrecht, The Netherlands

**Acknowledgements** This paper is part of the outputs emanating from the program entitled 'China National Health and Family Planning Commission and the Gates Foundation TB Project (Phase III)'—a collaboration between the Government of China and the Bill and Melinda Gates Foundation, and implemented by the China Center of Disease Control and Prevention (CDC). The authors of the paper also gratefully acknowledge the officers in Zhejiang Provincial CDC and the 11

prefectural CDCs for helping with collecting drug-resistant tuberculosis related policies in Zhejiang.

**Contributors** The study was designed by WJ, ST, CE and FC. FH, YP, XW, BC and WJ coordinated and conducted data collection. WJ conducted the literature review and wrote the manuscript as the first author. ST, CE, BC and FC provided suggestions on data analysis framework and data interpretation, and also revised the manuscript. All authors reviewed the draft manuscript, provided comments on the finalisation of the manuscript and have read and approved the manuscript in its current state.

**Funding** The work was supported by the Bill and Melinda Gates Foundation (grant number: OPP1149395).

**Competing interests** None declared.

**Patient consent for publication** Not required.

**Ethics approval** This study is under the overall evaluation study of China-Gates TB Project Phase III and used data collected from this project. The implementation of this project has received the ethical approval from China Center of Disease Control and Prevention (number: 201626). The protocol of the overall study design and data collection tools was reviewed and approved by the Institutional Review Board of Duke University (IRB approval code: 2017-0768). This paper does not use data involving human participants.

**Provenance and peer review** Not commissioned; externally peer reviewed.

**Data availability statement** Data may be obtained from a third party and are not publicly available. The data from the Tuberculosis Information Management System used in this study are owned by China CDC. The data could only be accessed after obtaining permission from China CDC.

**ORCID iD**
Weixi Jiang http://orcid.org/0000-0002-5643-6255

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
