## [Reviewer comments · BMJ Open]

ARTICLE DETAILS

TITLE (PROVISIONAL)	Policy changes and the screening, diagnosis and treatment of drug-resistant tuberculosis patients from 2015-2018 in Zhejiang Province, China: a retrospective cohort study
AUTHORS	Jiang, Weixi; Peng, Ying; Wang, Xiaomeng; Elbers, Chris; Tang, Shenglan; Huang, Fei; Chen, Bin; Cobelens, Frank

VERSION 1 – REVIEW

REVIEWER	Mahomed, Sharana Centre for the Aids Programme of Research in South Africa
REVIEW RETURNED	06-Jan-2021

GENERAL COMMENTS	Drug resistant tuberculosis remains a public health concern globally. This study evaluated programmatic performance of individuals in Zhejiang from 2015 to 2018. A cascade analysis approach was used and examined screening, diagnosis, treatment and management. This study is relevant globally as TB policies and guidelines have dramatically changed in all settings due to the advent of real time PCR assays. These changes have influenced case finding and treatment of DR-TB patients in diverse ways. Furthermore, this study also looked at the way these policy changes have influenced patients of socio-economic status. The authors describe a cascading analysis approach which will be beneficial to replicate in other settings in order to determine implementation strategies that increase case findings and allow for optimal management of infected patients.
--

REVIEWER	Bhavaraju, Rajita Rutgers University
REVIEW RETURNED	

GENERAL COMMENTS	Study is well done. A few questions, answers to which may make the paper more clear: 1-Policy changes were assessed by prefecture report. Was there a more objective easy to make this assessment? 2 - Were there any significant demographic changes in the prefectures that may have have affected rates of DRTB besides testing technology and including criteria?
--

REVIEWER	Haile, Sarah University of Zurich, Epidemiology, Biostatistics and Prevention Institute
REVIEW RETURNED	18-Feb-2021

GENERAL COMMENTS	Differences between prefectures in terms of both policy and results seem to be key here, as well as differences before and after policy changes within those prefectures. While I understand that it is not ideal to provide results of individual prefectures for political reasons, would it not be of interest to show results from groups of prefectures with similar policies? I find that information now in the model results, but it doesn't include changes to policies. Further, there are no graphical results by policy group. A description of and reference for cascade analysis would be very helpful in the methods section. The reference provided for it (9) in the introduction did not appear to be about the methodology. Were differences before / after policy changes explored? It would appear that a flow chart approach would be useful for visualizing the cascade of subjects here, and such figures are shown on the website https://cascade.tools/tool Would it be possible to incorporate this kind of figure here? What ethnicity categories were used? P-values in the text should be given to more precision. "P<0.05" is not acceptable. Please provide (a translation of) the policy questionnaire as supplementary material. A review of the language would be helpful. I found a number of typographical errors.
---

VERSION 1 – AUTHOR RESPONSE

Reviewer Reports:

Reviewer: 1

Dr. Sharana Mahomed, Centre for the Aids Programme of Research in South Africa

Comments to the Author:

Drug resistant tuberculosis remains a public health concern globally. This study evaluated programmatic performance of individuals in Zhejiang from 2015 to 2018. A cascade analysis approach was used and examined screening, diagnosis, treatment and management.

This study is relevant globally as TB policies and guidelines have dramatically changed in all settings due to the advent of real time PCR assays. These changes have influenced case finding and treatment of DR-TB patients in diverse ways. Furthermore, this study also looked at the way these policy changes have influenced patients of socio-economic status.

Thanks a lot for your comments. Our study hopes to provide some evidence on how real time PCR assays could be more effectively used in DRTB case finding, combined with other policies. To explore the potential differential impacts of policies on patients of different socio-economic status is also one of our study aims

The authors describe a cascading analysis approach which will be beneficial to replicate in other settings in order to determine implementation strategies that increase case findings and allow for optimal management of infected patients.

Thanks a lot for your comments. Our study does hope to provide a comprehensive analysis on the whole procedure of care for DRTB patients through cascade analysis, and find potential strategies to improvement the management of infected patients.

Reviewer: 2

Dr. Rajita Bhavaraju, Rutgers University

Comments to the Author:

Study is well done. A few questions, answers to which may make the paper more clear:

1 - Policy changes were assessed by prefecture report. Was there a more objective easy to make this assessment?

Thanks a lot for pointing this out, we did try to collect some policy documents, but the policies on different aspects of DRTB care, such as screening, reimbursement and financial assistance are issued by different government agencies, and some are not available. The TB departments in the CDCs are responsible for all TB related work and are familiar with all these policies, and that is why we chose to organize the questionnaire survey at the local prefecture-level CDC.

2 - Were there any significant demographic changes in the prefectures that may have affected rates of DRTB besides testing technology and including criteria?

Thanks a lot for pointing this out. We have checked the demographic statistics, that the resident population in Zhejiang was 55.4 million in 2015, and 57.4 million in 2018, basically the increase is around 5%. Nevertheless, the number of presumptive patients tested for DRTB in 2018 is almost 3 times the number in 2015 (18670 vs. 6463). We therefore think that in our study the influence of the demographic changes is minimal compared to policy changes, and we have added this explanation in the limitations of the discussion section.

Reviewer: 3

Dr. Sarah Haile, University of Zurich

Comments to the Author:

Differences between prefectures in terms of both policy and results seem to be key here, as well as differences before and after policy changes within those prefectures. While I understand that it is not ideal to provide results of individual prefectures for political reasons, would it not be of interest to show results from groups of prefectures with similar policies? I find that information now in the model results, but it doesn't include changes to policies. Further, there are no graphical results by policy group.

Thanks a lot for pointing this out. In fact there are changes in policies for each prefecture during 2015-2018, and the same type of change may have happened in different years for different prefectures, and policy changes may have happened multiple times for one prefecture. Therefore, we determined the policy combination category that applied to the patients in each prefecture in each year based on the policy survey questionnaire, and Figure 3 reflects policy changes in each prefecture overtime. In the regression models we explored the association between the policy combinations and DRTB patient registration and treatment after adjusting for other factors. However, it is difficult to show results from the groups of prefectures with similar policies due to the fact that multiple changes may have happened in the same prefecture. The graphical results regarding the correlation between policy combinations and DRTB registration/treatment performance are also difficult to present as simple variables. We therefore think that the regression result table represents and explains the results most clearly.

A description of and reference for cascade analysis would be very helpful in the methods section. The reference provided for it (9) in the introduction did not appear to be about the methodology.

Thanks a lot for pointing this out. The reference number on the cascade methodology should be 29, and we have revised this.

Were differences before / after policy changes explored?

Yes, as we mentioned in response to your first comment, the same type of change may have happened in different years for different prefectures, and the policy may have changed several times for one prefecture. Therefore, we determined the policy combination category that applied to the patients in each prefecture in each year based on the policy survey questionnaire, and included the policy combinations as a categorical variable in the regression analysis.

It would appear that a flow chart approach would be useful for visualizing the cascade of subjects here, and such figures are shown on the website [https://urldefense.com/v3/__https://cascade.tools/tool__;!!OToaGQ!4dO3WW9qJiRCtZuzUHVHpdIDhOqTHJELkGOjJncgvkVmigCC1d9Yon9guFgyt6VyimF9ad79gL0\\$](https://urldefense.com/v3/__https://cascade.tools/tool__;!!OToaGQ!4dO3WW9qJiRCtZuzUHVHpdIDhOqTHJELkGOjJncgvkVmigCC1d9Yon9guFgyt6VyimF9ad79gL0$) Would it be possible to incorporate this kind of figure here?

Thanks a lot for pointing this out. We did try to include a figure to show the cascade at the beginning, and the problem is that the patient attrition at early stages is huge. For example, in 2018 18670 had taken DST test but only 1580 were diagnosed as DRTB, and 663 as RR/MDRTB. If height of the histogram is produced in proportion to the numbers, the figure would look very imbalanced, and the information on the steps after diagnosis would be very vague on the figure. That is why we gave up using figures, and chose to use two tables for the stages before diagnosis and after diagnosis separately. Besides, we also hope to present the results for each year to show changes, that would need 4 figures, but the effects of presentation do not look better than a table.

What ethnicity categories were used?

We categorized the ethnicity of the study population as Han and other minority groups, and we have added this explanation in the method section. In Zhejiang, for the overall population, over 99% are Han People according to national statistics

P-values in the text should be given to more precision. "P<0.05" is not acceptable. Thanks a lot for pointing this out, and we have revised the text accordingly.

Please provide (a translation of) the policy questionnaire as supplementary material. Thanks a lot for pointing this out, and we have provided a translated copy of the policy questionnaire as supplementary material.

A review of the language would be helpful. I found a number of typographical errors. Thanks a lot for pointing this out. We have reviewed the paper carefully and made language edits.

VERSION 2 – REVIEW

REVIEWER	Bhavaraju, Rajita Rutgers University
REVIEW RETURNED	09-Mar-2021
GENERAL COMMENTS	not applicable
REVIEWER	Haile, Sarah University of Zurich, Epidemiology, Biostatistics and Prevention Institute
REVIEW RETURNED	06-Mar-2021

GENERAL COMMENTS	Thanks for your revision.
---------------------------